# Biomechanical Gait Analysis Using a Smartphone-Based Motion Capture System (OpenCap) in Patients with Neurological Disorders

**DOI:** 10.3390/bioengineering11090911

**Published:** 2024-09-12

**Authors:** Yu-Sun Min, Tae-Du Jung, Yang-Soo Lee, Yonghan Kwon, Hyung Joon Kim, Hee Chan Kim, Jung Chan Lee, Eunhee Park

**Affiliations:** 1Department of Rehabilitation Medicine, School of Medicine, Kyungpook National University, Daegu 41944, Republic of Korea; ssuni119@naver.com (Y.-S.M.); teeed0522@hanmail.net (T.-D.J.); leeyangsoo@knu.ac.kr (Y.-S.L.); 2Department of Rehabilitation Medicine, Kyungpook National University Chilgok Hospital, Daegu 41404, Republic of Korea; rlagudwns8851@naver.com; 3AI-Driven Convergence Software Education Research Program, Graduate School of Computer Science and Engineering, Kyungpook National University, Daegu 41566, Republic of Korea; 4Department of Biomedical Engineering, College of Medicine, Seoul National University, Seoul 03080, Republic of Korea; hckim@snu.ac.kr (H.C.K.); ljch@snu.ac.kr (J.C.L.); 5Department of Rehabilitation Medicine, Kyungpook National University Hospital, Daegu 41944, Republic of Korea; k55054103@gmail.com; 6Interdisciplinary Program in Bioengineering, Graduate School, Seoul National University, Seoul 08826, Republic of Korea; 7Institute of Bioengineering, Seoul National University, Seoul 03080, Republic of Korea; 8Institute of Medical and Biological Engineering, Medical Research Center, Seoul National University, Seoul 03080, Republic of Korea

**Keywords:** gait, kinematics, kinetics, motion capture, smartphone

## Abstract

This study evaluates the utility of OpenCap (v0.3), a smartphone-based motion capture system, for performing gait analysis in patients with neurological disorders. We compared kinematic and kinetic gait parameters between 10 healthy controls and 10 patients with neurological conditions, including stroke, Parkinson’s disease, and cerebral palsy. OpenCap captured 3D movement dynamics using two smartphones, with data processed through musculoskeletal modeling. The key findings indicate that the patient group exhibited significantly slower gait speeds (0.67 m/s vs. 1.10 m/s, *p* = 0.002), shorter stride lengths (0.81 m vs. 1.29 m, *p* = 0.001), and greater step length asymmetry (107.43% vs. 91.23%, *p* = 0.023) compared to the controls. Joint kinematic analysis revealed increased variability in pelvic tilt, hip flexion, knee extension, and ankle dorsiflexion throughout the gait cycle in patients, indicating impaired motor control and compensatory strategies. These results indicate that OpenCap can effectively identify significant gait differences, which may serve as valuable biomarkers for neurological disorders, thereby enhancing its utility in clinical settings where traditional motion capture systems are impractical. OpenCap has the potential to improve access to biomechanical assessments, thereby enabling better monitoring of gait abnormalities and informing therapeutic interventions for individuals with neurological disorders.

## 1. Introduction

Pediatric gait analysis is a critical tool for assessing motor function, particularly in children with neurological disorders such as cerebral palsy [1]. Accurate gait analysis is essential for designing appropriate therapeutic interventions, such as botulinum toxin injections and robotic-assisted gait training [2]. However, conducting traditional gait analysis in pediatric populations presents significant challenges [3]. Children, especially those with neurological impairments, often struggle with cooperation and adherence to the structured protocols required for accurate data collection. For instance, young children may find it difficult to follow instructions during gait assessment, leading to inconsistent and unreliable data. Moreover, the necessity for assistive devices like orthoses further complicates the process, as these devices can interfere with the natural gait pattern and thus affect the validity of the analysis.

These challenges underline the pressing need for simpler, more adaptable gait analysis tools that can accommodate the unique needs of pediatric populations. The traditional gait analysis setup, typically involving a gait laboratory equipped with multiple cameras, force plates, and motion capture systems, is not only resource-intensive but also requires substantial technical expertise and patient compliance [4]. As a result, there has been growing interest in developing alternative methods that are less burdensome and more accessible for use in clinical settings. Recent technological advancements have introduced innovative tools like smartphone-based motion capture systems, which offer a more practical solution without compromising on accuracy [5]. These systems, such as the OpenCap software, leverage the widespread availability and user-friendly nature of smartphones to perform 3D motion analysis and musculoskeletal force estimations with minimal setup [6]. This approach not only reduces the technical and logistical barriers associated with traditional gait analysis but also enhances the feasibility of conducting assessments in diverse clinical environments, including settings where full-scale gait labs are not available. Furthermore, the use of these accessible technologies can facilitate more frequent and comprehensive assessments, allowing clinicians to monitor patient progress more effectively and adjust treatment plans in a timely manner. As these tools continue to evolve, they hold the potential to revolutionize gait analysis by making it more scalable and adaptable to a broader range of clinical applications.

The development of such tools represents a significant step forward in pediatric gait analysis, particularly for children with cerebral palsy and other neurological conditions who may benefit from more frequent and accessible evaluations. For instance, studies have shown that regular gait assessments in children with cerebral palsy can lead to more timely interventions, improving mobility and overall quality of life [3,7,8]. Similarly, in stroke and Parkinson’s disease patients, frequent gait analysis using accessible tools has been shown to help in the early detection of gait deterioration, enabling prompt adjustments in therapeutic strategies [7,8,9,10,11]. These advancements not only improve individualized care but also support large-scale studies that can further our understanding of gait abnormalities across different neurological conditions. As gait analysis plays a vital role in tracking the progress of therapeutic interventions, the adoption of simpler, yet reliable, methods is crucial for improving patient outcomes. Moreover, these innovations could facilitate large-scale studies and clinical trials by making gait analysis more accessible and scalable, thus contributing to the broader field of pediatric rehabilitation and biomechanics.

Quantitative gait analysis has historically been utilized primarily within limited, small-scale groups due to the high costs, need for specialized personnel, and the significant time and resource requirements involved [10]. This type of analysis demands the installation and operation of expensive equipment, such as 3D motion capture systems, force plates, and electromyography devices, all of which require skilled professionals trained in their use. Additionally, the data collection and analysis processes are complex and time-consuming, posing substantial economic challenges for large-scale studies or clinical applications. These factors have hindered the widespread adoption of gait analysis, resulting in its predominant use within specific research groups or advanced medical institutions. Recent advances in computer vision have made it possible to perform quantitative movement analysis using only digital videos from low-cost devices like smartphones and tablets. These technologies, such as OpenPose, automatically identify keypoints on the human body from simple videos, significantly improving the accessibility of movement assessment [5,12,13]. This has led to the development of various workflows for measuring gait parameters and joint kinematics, demonstrating the potential for widespread application in health and performance analysis. Building on these advancements, the development of user-friendly software has further enhanced the accessibility of these analytical tools, paving the way for broader dissemination and utilization in clinical and research settings in the future.

Recently, OpenCap, an open-source, web-based software, has been developed to enable the estimation of human 3D kinematics and dynamics from videos captured using two or more smartphones [6]. This software is freely available and leverages decades of advancements in computer vision and musculoskeletal simulation, allowing for the analysis of movement dynamics without the need for specialized hardware, software, or expertise. However, while OpenCap has been validated on musculoskeletal models of healthy individuals, its application to patients with clinical conditions has yet to be explored. To address this gap, we have designed a study to investigate whether this accessible and user-friendly technology can be effectively utilized in clinical settings with actual patients, thus extending its utility beyond the laboratory and into practical healthcare environments.

This study advances the field of gait analysis by applying the OpenCap smartphone-based motion capture system in clinical settings to assess patients with neurological disorders such as stroke, Parkinson’s disease, and cerebral palsy. We demonstrate the system’s capability to capture significant kinematic and kinetic differences between healthy controls and patients, showcasing its effectiveness in clinical environments. Additionally, this research highlights the potential of low-cost, accessible technology to perform complex gait analyses, thereby expanding the feasibility of biomechanical assessments in diverse settings and supporting more frequent and comprehensive monitoring of gait abnormalities in patients.

## 2. Materials and Methods

### 2.1. Participants and Sample Size Estimation

The experiment involved two groups: a patient group comprising 10 individuals with neurological disorders, including stroke, Parkinson’s disease, cerebral palsy, and others, and a control group of 10 healthy adults. The sample size was determined based on the prior literature and statistical power considerations, aiming to achieve 80% power with a large, anticipated effect size (d = 0.8). Functional Data Analysis (FDA) was conducted, and bootstrap simulations were used to estimate confidence intervals for key outcome measures [14,15,16]. In some instances, the confidence intervals between the patient and control groups did not overlap, indicating that the sample size was sufficient to detect statistically significant differences. These results suggest that the current sample size was adequate for identifying substantial group differences, validating the use of FDA and bootstrap methods. All participants were fully informed of the study’s procedures and provided written consent prior to data collection. The study was approved by the ethics review committee of KNUCH (Protocol Code: 2023-09-031).

### 2.2. Experimental Protocol and Equipment

The experimental setup was conducted in accordance with the guidelines provided by the OpenCap software and hardware configuration protocol (Appendix A) [6].

#### 2.2.1. Camera Setup and Calibration 

Two iOS devices with rear-facing cameras were used to capture movement while minimizing segment occlusions by positioning the cameras 30–45° off the participant’s forward-facing line. This setup was chosen to avoid limb occlusion often caused by purely sagittal views. Participants were first observed to ensure full visibility throughout the task, and background distractions were minimized. Calibration was performed using a checkerboard (210 × 175 mm) printed on A4 paper, with accurate dimensional verification. The checkerboard was placed within the camera’s field of view to ensure correct calibration of extrinsic parameters, maintaining visibility from all cameras within a 5 m distance to achieve precise data collection.

#### 2.2.2. Video Collection and Pose Estimation

Following calibration, videos were recorded at a resolution of 720 × 1280 pixels and a frame rate of 60 Hz. Pose estimation was performed using two algorithms, OpenPose and High-Resolution Network (HRNet), with 20 body keypoints selected for further analysis, including the neck, mid-hip, shoulders, hips, knees, ankles, heels, toes, elbows, and wrists. OpenCap uses a Direct Linear Transformation algorithm to triangulate 3D keypoint positions from synchronized 2D video data, weighted by keypoint confidence scores [12,17,18,19,20]. However, because 3D keypoints from video alone are insufficient for full biomechanical analysis, two LSTM networks were trained to predict the positions of 43 anatomical markers from 20 triangulated video keypoints, enhancing the robustness of joint kinematics.

#### 2.2.3. Physics-Based Modeling and Simulation 

After calibration, OpenCap guides users to record the participant in a neutral standing pose, which is used to scale a musculoskeletal model to the participant’s anthropometry via OpenSim’s Scale tool. OpenCap utilizes the musculoskeletal model developed by Lai et al. [21,22], incorporating modifications to the hip abductor muscle paths as described by Uhlrich et al. [23]. The musculoskeletal model, consisting of 33 degrees of freedom across the pelvis (6 DOF in the ground frame), hips (3 DOF each), knees (1 DOF each), ankles (2 DOF each), metatarsophalangeal joints (1 DOF each), lumbar region (3 DOF), shoulders (3 DOF each), and elbows (2 DOF each), is driven by 80 muscles and 13 ideal torque motors, with ground reaction forces simulated through foot–ground contact spheres. Once scaled, OpenCap computes joint kinematics using OpenSim’s Inverse Kinematics tool, with users able to visualize the resulting 3D kinematics within the web application.

### 2.3. Experimental Procedure

Participants were instructed to walk at a self-selected speed along a flat 4 m path after achieving the static posture required by the equipment and establishing a connection with the camera system. Each participant completed the walk three times, and the best performance was selected for analysis. For participants who were unable to walk independently or required supervision, a support individual was positioned behind them to ensure safety while allowing the patient to appear as large as possible within the camera frame.

### 2.4. Data Collection and Processing

In this study, coordinate data collected from experimental marker trajectories were exported as Track Row Column (TRC) files and analyzed using the Inverse Kinematics tool in OpenSim 4.5 (SimTK, Stanford University, Stanford, CA, USA) [24]. Temporal variations in the kinematics of the pelvis, hip, knee, and ankle were examined throughout the gait cycle. Subsequently, the Inverse Dynamics tool was employed to calculate joint forces, offering insights into the time-dependent changes in joint kinetics. Importantly, the inverse dynamics analysis was performed under the explicit assumption of an external force-free condition. This approach enabled us to isolate the internal joint moments and forces generated by the system itself, thereby providing a more focused investigation into the intrinsic dynamics of movement.

### 2.5. Statistical Analysis 

The differences in baseline characteristics between the patient and control groups were statistically evaluated. Continuous variables were analyzed using the Mann–Whitney U test, while categorical variables were assessed using Fisher’s Exact test. A *p*-value of less than 0.05 was considered statistically significant. Functional Data Analysis (FDA) was applied to time-series kinematic and kinetic data for key gait parameters (pelvis, hip, knee, and ankle) to model continuous joint movements across the entire gait cycle [14,25,26,27]. Each subject’s gait cycle was normalized to 100% to facilitate comparisons between the patient and control groups. Bootstrap resampling with 1000 iterations was used to generate 95% confidence intervals (CIs) for each group, assessing the variability and statistical significance of joint kinematics and kinetics [15,16]. Non-overlapping regions of the CIs between the two groups were identified as statistically significant, highlighting phases of the gait cycle where kinematic and kinetic deviations occurred. All analyses were performed using custom scripts in Python 3.11.4, employing statistical libraries such as statsmodels, scipy, numpy, and matplotlib for resampling and visualization, ensuring a robust comparison of the time-series data.

## 3. Results

### 3.1. Patient Characteristics

The demographic and clinical characteristics of the control and patient groups are presented in Table 1. The patient group consisted of four individuals diagnosed with stroke, two with Parkinson’s disease, and four with other neurological disorders, including cerebral palsy, Guillain–Barré syndrome, and cerebellar ataxia. The patient group was significantly older (mean age: 51.60 years) compared to the control group (mean age: 31.30 years; *p* = 0.034). The sex distribution was similar between the groups, with four males and six females in the patient group, and three males and seven females in the control group (*p* = 1.000). No significant differences were observed between the groups in terms of height, weight, BMI, or cadence. However, the patient group exhibited a significantly slower gait speed (*p* = 0.002), shorter stride length (*p* = 0.001), and greater step length asymmetry (*p* = 0.023). Additionally, the patient group demonstrated a significantly wider step width compared to the control group (*p* = 0.045).

### 3.2. Kinematic Findings during the Gait Cycle in the Control Group 

Figure 1 illustrates the kinematic data of 10 healthy individuals during a gait cycle, representing the angular movement of key body segments across nine distinct parameters: pelvic tilt, pelvic list, pelvic rotation, hip flexion/extension, hip adduction/abduction, hip internal/external rotation, knee flexion/extension, ankle dorsiflexion/plantarflexion, and subtalar inversion/eversion. Each graph plots the average angular movement across the gait cycle, with the blue line representing the mean values and the shaded region indicating the standard deviation (±1 SD). The data reveal that pelvic tilt oscillates within a range of approximately 5°, while pelvic list demonstrates lateral shifting within ±5°, and pelvic rotation fluctuates up to ±10°. Hip flexion/extension exhibits the most significant angular variation, spanning from approximately −20° to 30°, with a marked flexion phase during the early stance followed by extension. Hip adduction/abduction oscillates around ±10°, while internal/external hip rotation remains relatively stable within ±10°. Knee flexion peaks at around 70° during the early stance, while ankle dorsiflexion/plantarflexion ranges between −10° and 10°, reflecting typical ankle movement during gait. Lastly, subtalar inversion/eversion exhibits subtle oscillations within ±10°, consistent with normal foot movement during walking. These kinematic patterns collectively represent the typical biomechanical behavior of healthy individuals during gait, providing a robust reference for analyzing deviations in pathological gait conditions.

### 3.3. Kinematic Findings during the Gait Cycle in the Patient Group 

The kinematic data of the 10 patients during the gait cycle reveal notable deviations from normal gait mechanics, characterized by increased variability and reduced control across multiple joint movements (Figure 2). Pelvic tilt fluctuates between −10° and 5°, with standard deviations reaching ±15°, while pelvic list shows exaggerated lateral shifts between −10° and 10°, especially in the mid to late gait phase. Pelvic rotation exhibits similar angular ranges as healthy controls but with greater variability, particularly towards the latter half of the cycle. Hip flexion/extension displays a constrained range of −15° to 30°, with reduced extension during the late stance, and hip adduction/abduction shows diminished adduction with fluctuations from −5° to 10°. Internal/external rotation of the hip shows increased instability, with deviations up to ±12°, particularly during the swing phases. Knee flexion reaches similar peaks as normal controls, but the variability is heightened during the early stance, while ankle dorsiflexion/plantarflexion demonstrates diminished dorsiflexion control, particularly during the stance phase, with deviations reaching ±10°. Subtalar inversion/eversion fluctuates more widely between −5° and 10°, with increased variability of up to ±8°, suggesting instability in foot mechanics. These data indicate that the patients exhibited increased variability and less consistent joint control, particularly during key phases of the gait cycle, reflecting diverse compensatory strategies.

### 3.4. Kinematic Differences during the Gait Cycle: A Comparison between Control and Patient Groups 

The comparison of gait kinematics between the patient and control groups revealed significant deviations across multiple joints, particularly in pelvic tilt, hip rotation, knee flexion, and ankle dorsiflexion during key phases of the gait cycle (Figure 3). For instance, in the pelvic tilt, there are substantial non-overlapping regions between 0–20% and 60–80% of the gait cycle, with patients showing increased variability and larger tilt (ranging from −10° to 10°) compared to controls (typically within −5° to 5°). Similarly, in hip flexion/extension, a significant difference is observed between 50 and 60% of the gait cycle, where patients demonstrate reduced extension, peaking at 20°, compared to controls, who achieve near 30°. The knee flexion/extension also reveals significant divergence around 20–30% of the gait cycle, where patients show reduced flexion (approximately 50°) compared to the control group (around 70°). These statistically significant differences, marked by non-overlapping CIs, underscore the altered gait mechanics in patients, reflecting biomechanical impairments, reduced mobility, and potential compensatory strategies during gait.

Table 2 summarizes the peak joint angles for the hip, knee, ankle, and subtalar joints in both the control and patient groups, including their respective means and standard deviations (SD). The comparison of the maximum, minimum, and mean values of the kinematic parameters across the gait cycle is presented in Appendix A. A significant difference was observed in hip extension (*p* = 0.007), indicating a marked discrepancy between the control and patient groups in this movement. No statistically significant differences were found for the other joint movements.

### 3.5. Gait Cycle Kinematics in Stroke Patients versus Controls: A Subgroup Comparison 

Figure 4 compares the kinematic gait data of 10 healthy individuals and 10 stroke patients, focusing on key parameters such as the pelvis, hip, knee, and ankle joints. “In pelvic tilt, there are substantial non-overlapping regions between 0–40% and 50–60% of the gait cycle, indicating significant differences in tilt control, with stroke patients showing greater fluctuations, tilting from −2.5° to 5°, compared to the controls, whose tilt remained more stable around ±2°. The pelvic list shows less pronounced variability between groups, although the stroke group exhibits a broader fluctuation, especially during the mid-phase of the gait cycle. In pelvic rotation, significant differences emerge towards the end of the gait cycle (90–100%), where stroke patients show reduced rotation control, with ranges from 5° to −7.5°, while controls maintain a more steady pattern between 5° and −5°. For hip flexion/extension, significant differences occur around 40–60% of the gait cycle, where stroke patients display a more restricted range of motion (hip flexion peaking at 20° compared to the control group at nearly 30°). In hip adduction/abduction, notable deviations are observed near the end of the gait cycle (80–100%), where stroke patients demonstrate reduced adduction capacity, with values around −5° to −7.5° compared to controls. The hip internal/external rotation shows increased variability across the entire cycle for stroke patients, though without marked regions of non-overlapping significance. The knee flexion/extension reveals significant differences between 40 and 60% of the gait cycle, where stroke patients exhibit less flexion (around 40–50°) compared to the control group (approaching 70°). In ankle dorsiflexion/plantarflexion, non-overlapping regions are present at both the early and late stages of the gait cycle (0–20% and 80–100%), indicating substantial impairments in stroke patients, with their dorsiflexion being markedly reduced, ranging between −5° and 5° compared to the broader range of −10° to 10° seen in controls. Lastly, in subtalar inversion/eversion, although both groups exhibit similar trends, the stroke patients display greater variability, particularly in the mid-stance phase, with more pronounced deviations around inversion control. These differences in kinematic patterns suggest significant impairments in joint mobility and stability in stroke patients, especially in pelvic control, hip and knee flexion/extension, and ankle motion. The regions of non-overlapping confidence intervals underline the severity of gait deviations in stroke patients, likely due to neurological deficits that affect motor control and coordination during ambulation.

### 3.6. Gait Cycle Kinematics in Parkinson’s Disease Patients versus Healthy Controls: A Subgroup Comparison 

The comparison of gait kinematics between Parkinson’s disease patients and controls revealed significant deviations across multiple joints during key phases of the gait cycle (Figure 5). Notably, pelvic tilt does not exhibit any statistically significant differences between the groups, as evidenced by the absence of red-highlighted regions, indicating that both groups maintain similar pelvic tilt patterns throughout the gait cycle. However, in pelvic list, significant differences are observed between 40 and 50% of the gait cycle, with the Parkinson’s group displaying greater lateral deviations compared to controls. Pelvic rotation shows a statistically significant difference around 60% of the gait cycle, with the Parkinson’s group exhibiting reduced rotational control. Differences are also prominent in hip flexion/extension between 60 and 75% of the gait cycle, where Parkinson’s patients show reduced hip extension. In hip adduction/abduction, significant deviations occur around 40–60% of the cycle, indicating reduced adduction in the Parkinson’s group. For hip internal/external rotation, there are significant differences at 50–60% and 90–100% of the cycle, reflecting greater variability in the Parkinson’s group. Similarly, knee flexion/extension differs significantly between 50 and 60%, with reduced flexion in Parkinson’s patients. Ankle dorsiflexion/plantarflexion shows significant deviations at three phases of the gait cycle—70% and 80–90%—with Parkinson’s patients demonstrating reduced dorsiflexion control. Lastly, subtalar inversion/eversion presents statistically significant differences in early phases of the cycle, indicating greater instability in foot mechanics among Parkinson’s patients. These findings suggest that individuals with Parkinson’s disease experience significant motor impairments across multiple joints during walking, with increased variability and reduced control particularly evident in hip, knee, and ankle movements.

### 3.7. Gait Cycle Kinematics in Pediatric Patients versus Healthy Controls: A Subgroup Comparison

Figure 6 compares the kinematic gait data of pediatric patients and healthy controls across key parameters such as the pelvis, hip, knee, and ankle joints. In pelvic tilt, significant differences are observed across 0–90% of the gait cycle, with pediatric patients exhibiting greater fluctuations, ranging from −30° to 0°, compared to controls, whose tilt remains more stable within −5° to 5°. In pelvic list, significant differences are observed around 0–10%, 50–60%, and 80–100% of the gait cycle, with pediatric patients showing greater lateral deviations (−15° to 5°) compared to controls (−5° to 5°). For pelvic rotation, significant differences occur within the 0–30% range, with pediatric patients displaying increased variability, fluctuating between 0° and 30°, while controls remain more stable at −5° to 5°. In hip flexion/extension, differences are noted around 0–10%, 40–50%, and 90–100% of the gait cycle, where pediatric patients exhibit higher peak hip flexion (around 50°) compared to controls (30–40°). In knee flexion/extension, significant differences appear around 0–20%, 70–80%, and 90–100% of the gait cycle, with pediatric patients showing reduced knee flexion, peaking at 50°, compared to 70° in controls. In ankle dorsiflexion/plantarflexion, differences are evident across a large portion of the gait cycle, specifically 0–10% and 20–100%, with pediatric patients showing restricted movement between −30° and 5°, compared to controls who range from −20° to 15°. Lastly, in subtalar inversion/eversion, significant differences are observed between 20–50% and 80–100% of the gait cycle, where pediatric patients exhibit greater variability, with values ranging from −5° to 20°, compared to controls whose range is between −10° and 10°. These findings indicate that pediatric patients exhibit significant motor control differences across multiple joints, characterized by greater variability and reduced stability during gait, particularly in the hip, knee, ankle, and subtalar joints

### 3.8. Kinetic Differences during the Gait Cycle: A Comparison between Control and Patient Groups

The analysis of joint moments during the gait cycle, normalized to body weight, demonstrated significant differences between the control and patient groups across the hip, knee, and ankle joints (Appendix A). In hip flexion/extension moments, the patient group exhibited increased variability, particularly during the mid-stance phase (30–35% of the gait cycle), with mean differences of up to 0.1 Nm/kg compared to the control group. Statistically significant differences were observed, particularly in mid-stance and the terminal stance, as reflected by non-overlapping 95% confidence intervals. In knee flexion/extension moments, the patient group showed deviations of up to 0.05 Nm/kg during the loading response (8–10%) and mid-stance (30–35%) phases, indicating altered joint loading patterns. Finally, in ankle dorsiflexion/plantarflexion moments, the patient group demonstrated reduced plantarflexion moments during the push-off phase (40–45%), with differences reaching 0.05 Nm/kg compared to the control group. These deviations were statistically significant, as the 95% confidence intervals did not overlap with zero. Collectively, these findings suggest that patients with neurological disorders display altered joint moment patterns during gait, likely due to compensatory mechanisms and muscle impairments, particularly during critical phases of the gait cycle.

## 4. Discussion

This study demonstrated that OpenCap, an innovative open-source program, is highly effective in the simple and efficient measurement of kinematics and kinetics in both healthy individuals and patient populations. OpenCap’s ability to accurately capture movement dynamics using only video data from smartphones made it particularly useful in clinical settings where traditional motion capture systems are either impractical or unavailable. The software’s user-friendly interface and robust algorithms enabled the collection of comprehensive gait data, which were successfully applied to various patient groups, including those with neurological conditions such as stroke, Parkinson’s disease, and cerebral palsy. The program’s adaptability to different patient needs, even in cases of impaired mobility, highlights its potential as a valuable tool for clinical gait analysis and for tracking patient progress over time.

### 4.1. Comparison of Temporospatial Gait Parameters and Efficiency of Data Acquisition in Neurological Conditions

The temporospatial gait parameters observed in our patient group closely align with findings reported in studies involving individuals with neurological conditions such as stroke, Parkinson’s disease, and cerebral palsy. For instance, our patient group’s mean gait speed was 0.67 m/s, which is consistent with the reduced gait speeds typically reported in stroke patients (0.4–0.8 m/s) and Parkinson’s disease patients (0.6–0.9 m/s) [7,9,11,28,29,30,31,32]. Similarly, the stride length in our patient group averaged 0.81 m, which is markedly shorter than the control group’s 1.29 m, and is comparable to the reduced stride lengths observed in patients with cerebral palsy (0.7–1.0 m) and Parkinson’s disease (0.8–1.1 m) [4,8,33,34]. Additionally, the increased step length asymmetry in our patient group (107.43%) is in line with findings from stroke and cerebral palsy patients, where asymmetry percentages typically exceed 100%, reflecting the challenges in maintaining symmetrical gait patterns in these populations.

These comparisons underscore the validity of our findings and suggest that the gait characteristics of our patient group mirror those seen in individuals with well-documented neurological impairments. Moreover, the data acquisition process in our study was notably efficient, with all measurements being completed within a brief period of under 10 min. This efficiency stands in stark contrast to traditional gait analysis methods, which often require extensive setup and prolonged data collection times. The use of iOS-based motion capture technology in our study allowed for rapid, yet accurate, acquisition of gait parameters, demonstrating not only the practicality of this approach but also its potential for broad clinical application. The ability to quickly and accurately assess gait characteristics in patients, especially those with neurological conditions, is critical for timely diagnosis and intervention, and our method offers a valuable tool for achieving this in diverse clinical settings.

### 4.2. Comparative Analysis of Kinematic Gait Patterns in Controls and Patients with Neurological Disorders

This study provided a comparative analysis of gait patterns between individuals with neurological conditions, such as stroke and Parkinson’s disease, and healthy controls. Our findings highlighted several key deviations in gait kinematics among neurological patients, consistent with prior studies using traditional marker-based systems [7,8,9]. In stroke patients, we observed a reduction in hip flexion during the swing phase, where the hip flexion decreased by up to 10 degrees compared to the normal range of 35 to 40 degrees [7,28,35]. This reduction contributes to the asymmetric gait patterns commonly observed in stroke patients, leading to increased joint loading on the unaffected side, as previously documented in studies that observed a 15–20% increase in joint loading on the contralateral side [29]. 

In comparison to previous studies on Parkinson’s disease (PD) gait patterns, the results from this study reinforce and add specificity to our understanding of motor impairments during walking. The reduction in hip extension observed between 60 and 75% of the gait cycle in this study aligns with prior research that has quantified reduced hip extension by up to 9–11 degrees in PD patients compared to healthy controls [36]. Similarly, ankle dorsiflexion impairments were prominent in this study, particularly between 70 and 90% of the gait cycle, reflecting the findings by Nanhoe-Mahabier et al., who reported that PD patients exhibit reduced dorsiflexion by approximately 6–8 degrees during the terminal stance phase [30]. Moreover, the increased variability in hip internal/external rotation and subtalar inversion/eversion at different phases of the gait cycle observed here aligns with earlier work by Frenkel-Toledo et al., who found increased variability in gait metrics, such as step length and time, and irregular joint movements [31]. Their research indicated that this variability is 20–30% higher in PD patients than in controls, contributing to balance instability. Finally, the deviations in pelvic list and rotation reported in the current study echo findings by Nieuwboer et al., which showed that PD patients have a 3–5 degree greater lateral pelvic list during the gait cycle, affecting overall gait symmetry [32]. By quantifying the significant differences across multiple joints, this study corroborates the extent of movement impairment in Parkinson’s disease, while these numbers help specify how PD patients deviate from normal gait patterns.

Despite the overall consistency with previous studies, there are some discrepancies, particularly in the estimation of joint forces and muscle activation patterns. Traditional gait analyses that incorporate force plates provide more precise measurements of ground reaction forces (GRF), which are crucial for accurate kinetic calculations. In our study, the absence of direct GRF measurements may have introduced some degree of inaccuracy in the joint force estimates, particularly during dynamic phases of the gait cycle. Studies using marker-based systems with GRF have been able to produce more refined joint force data, which may explain some of the differences observed [13].

### 4.3. Comparative Analysis of Kinematic Gait Patterns between Healthy Controls and Pediatric Patients with Neurological Disorders

In comparing pediatric patients with neurological conditions, such as cerebral palsy, to healthy controls, we found consistent deviations in gait kinematics, echoing results from previous research. Specifically, pediatric patients exhibited reduced hip extension by 5–10 degrees during the late stance phase. This reduction aligns closely with findings from studies on children with cerebral palsy, which reported hip extension reductions of up to 12 degrees [37]. Additionally, our study showed a reduction of 10–15 degrees in knee flexion during the swing phase, which closely matches the reductions reported in children with spastic diplegia, where knee flexion was reduced by 10–14 degrees. These reductions in knee flexion contribute to difficulties in limb clearance during gait, a common issue in pediatric neurological populations that often leads to a characteristic “stiff-knee” gait pattern [38]. Ankle kinematics in pediatric patients revealed restricted dorsiflexion during the early stance (10–20%) and push-off, with reductions of 10 degrees. This is in agreement with prior studies that reported a 6–8 degree reduction in dorsiflexion in children with cerebral palsy [33]. Restricted dorsiflexion limits the ability to achieve a smooth heel-to-toe transition during walking, which can negatively affect the efficiency and fluidity of gait. Collectively, these deviations highlight the significant impact that neurological impairments have on gait stability, range of motion, and overall mobility. These deviations likely result from compensatory mechanisms related to weakened musculature and poor neuromotor control, as previously suggested in studies [3,4,39]. Addressing these specific deviations through targeted rehabilitation programs, particularly those aimed at improving pelvic stability, joint mobility, and neuromuscular coordination, could lead to significant improvements in functional gait for pediatric patients. 

### 4.4. Limitations of the Study

One significant limitation of this study is the absence of direct ground reaction force (GRF) measurements, which are crucial for accurate kinetic analysis. GRF data provide essential insights into the external forces acting on the body during movement, and without these measurements, estimates of joint forces and muscle activations may lack precision, particularly during dynamic phases of the gait cycle [40,41]. The assumption of an external force-free condition in the inverse dynamics analysis further departs from real-world conditions, especially during the stance phase, where GRF plays a substantial role in influencing joint kinetics. As a result, the subtle yet important variations typically observed between these phases were less pronounced, complicating the interpretation of the data. Future studies could address this by integrating OpenCap with portable force plates or in-shoe sensors to capture GRF during movement [42]. Additionally, incorporating muscle-driven simulations with algorithmic differentiation would allow for more accurate estimations of joint dynamics by accounting for muscle activations, skeletal dynamics, and external forces [43]. Leveraging such approaches, supported by optimal control methods and tools like CasADi, would significantly enhance the accuracy of dynamic analyses, particularly during the load-bearing phases of movement [44].

While the camera-based motion capture system used in this study offers flexibility and ease of use, such markerless systems generally exhibit reduced accuracy compared to traditional marker-based systems, with factors such as occlusion, lighting, and camera angles potentially introducing errors in kinematic data, especially during complex or abnormal gait patterns [6,45,46,47]. Future research should include a validation of the accuracy of the OpenCap system in specific patient populations, such as those with stroke, Parkinson’s disease, and pediatric patients. Currently, the default model provided by OpenCap, the LaiUlrich2022 model, is a modified version of the Rajagopal model based on adult data [6,21]. Incorporating a pediatric-specific model, which accounts for differences in height, weight, and muscle proportions, could improve the accuracy of kinematic and dynamic estimations for pediatric populations, providing more reliable information across diverse age groups and clinical conditions.

Before initiating the gait analysis using the OpenCap system, a short period of static posture—either neutral standing or any pose—must be maintained for model scaling, typically within 5 s. This process requires the individual to be captured from two camera angles while maintaining a static position unaided, which poses challenges for patients with compromised balance or pediatric populations. Furthermore, many patients undergoing rehabilitation are unable to walk independently and rely on assistive devices such as canes or walkers, rendering them unable to undergo proper evaluation due to camera vision occlusion. Even for patients who do not use assistive devices but require close supervision to prevent falls, the lack of multi-person identification support in the OpenCap system presents difficulties in ensuring accurate measurements. In the future, the development of protocols that support multi-camera setups capable of tracking markers from the rear or software that allows for multi-person identification could greatly expand the usability of the OpenCap system for a broader range of patients, including those with neurological conditions who utilize assistive devices.

The relatively small sample size also limits the generalizability of our findings. With only 20 participants across different patient groups, the study may not fully represent the broader population of individuals with neurological impairments. A larger sample size would provide more statistical power, allowing for a more nuanced analysis and increased confidence in detecting subtle differences between the patient groups.

## 5. Conclusions

This study demonstrates that OpenCap, a smartphone-based motion capture system, is a feasible and cost-effective tool for clinical gait analysis in patients with neurological disorders. The system was able to capture significant differences in gait parameters between healthy controls and patients with conditions such as stroke, Parkinson’s disease, and pediatric patients. These findings highlight the potential of OpenCap to enhance accessibility to biomechanical assessments, offering a practical alternative for gait analysis in clinical settings where traditional motion capture systems may not be viable.

## Figures and Tables

**Figure 1 bioengineering-11-00911-f001:**
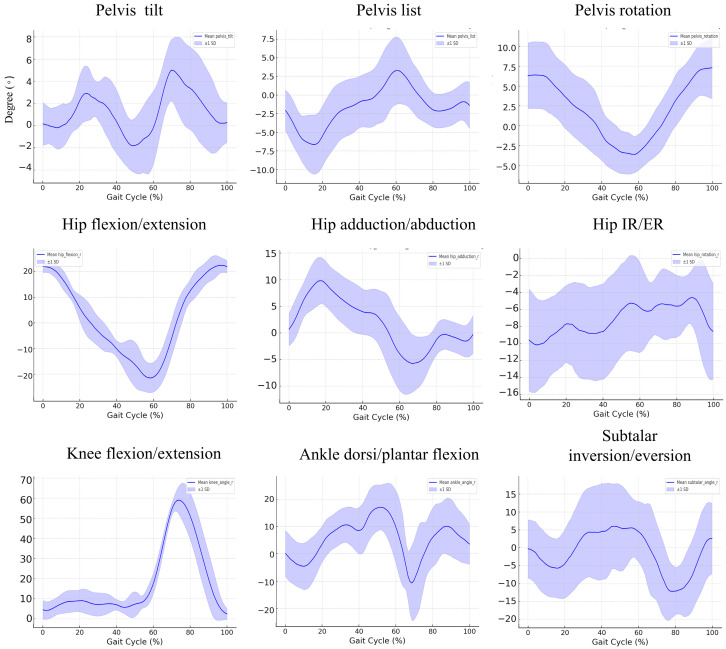
Joint-specific kinematic parameters during the gait cycle, normalized for a group of controls. Each subplot represents a specific joint movement across the gait cycle (%). The blue line indicates the mean kinematic angle, with shaded areas representing ±1 standard deviation (SD). The following movements are shown: pelvic tilt, list, and rotation; hip flexion/extension, adduction/abduction, and internal/external rotation (IR/ER); knee flexion/extension; ankle dorsiflexion/plantarflexion; and subtalar inversion/eversion.

**Figure 2 bioengineering-11-00911-f002:**
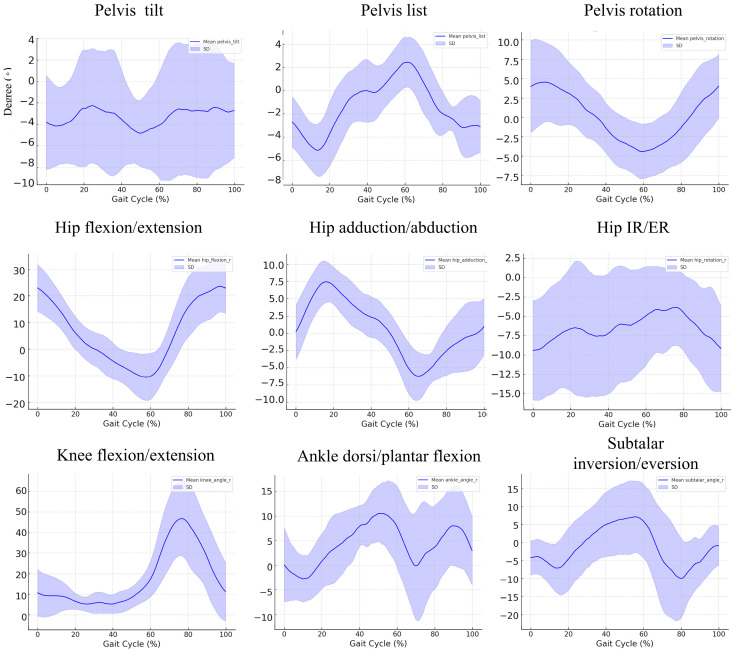
Joint-specific kinematic parameters during the gait cycle, normalized for a group of patients. Each subplot represents a specific joint movement across the gait cycle (%). The blue line indicates the mean kinematic angle, with shaded areas representing ±1 standard deviation (SD). The following movements are shown: pelvic tilt, list, and rotation; hip flexion/extension, adduction/abduction, and internal/external rotation (IR/ER); knee flexion/extension; ankle dorsiflexion/plantarflexion; and subtalar inversion/eversion.

**Figure 3 bioengineering-11-00911-f003:**
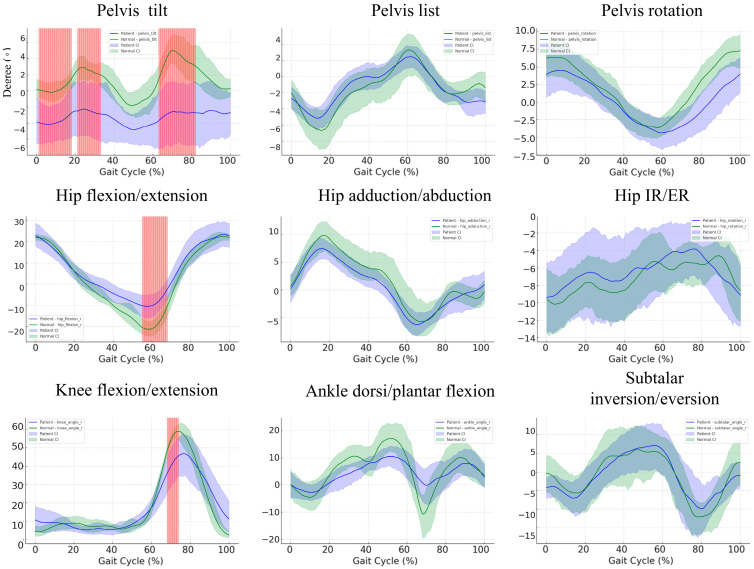
Bootstrap confidence bands and non-overlapping regions for hip, knee, and ankle parameters in patients and controls. This figure illustrates the bootstrap confidence intervals (CI) and regions of statistically significant differences between patients (blue) and controls (green) for key gait parameters. The shaded regions around each curve represent the 95% confidence intervals generated through 1000 bootstrap resamples. Red-highlighted vertical spans indicate areas where the confidence intervals of the two groups do not overlap, suggesting statistically significant differences in these regions.

**Figure 4 bioengineering-11-00911-f004:**
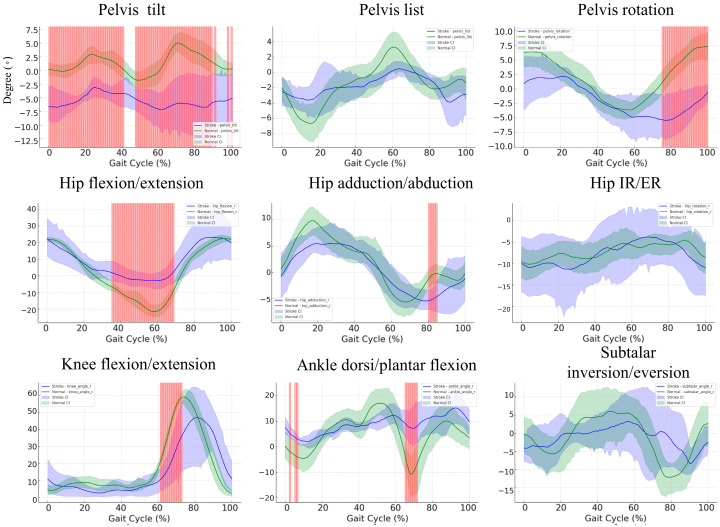
Comparison of bootstrap confidence intervals and non-overlapping regions for hip flexion, hip adduction, knee angle, and ankle angle in stroke patients and normal controls. The blue and green lines represent the mean curves for stroke patients and healthy controls, respectively, while the shaded regions indicate the 95% bootstrap confidence intervals (CI) calculated across the gait cycle. Red-highlighted regions indicate statistically significant differences where the confidence intervals between the two groups do not overlap.

**Figure 5 bioengineering-11-00911-f005:**
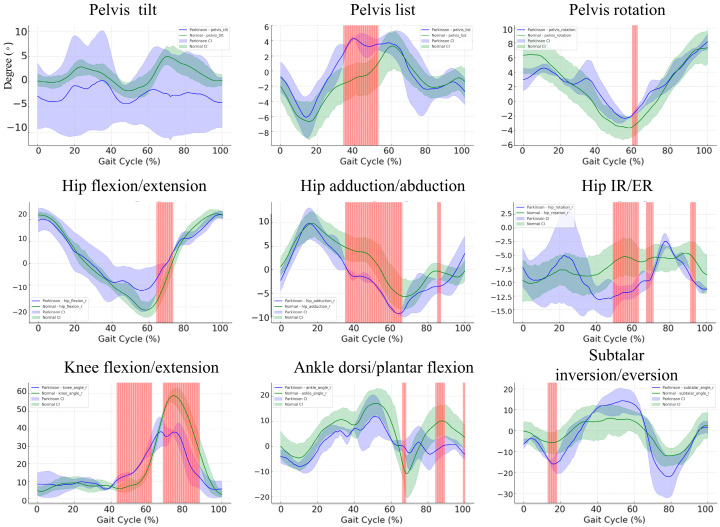
Comparison of bootstrap confidence intervals and non-overlapping regions for hip flexion, hip adduction, knee flexion, and ankle plantarflexion/dorsiflexion in the Parkinson’s patients and controls. The mean joint angle trajectories for Parkinson’s patients are shown in blue, while the green lines represent healthy controls. The shaded regions around each curve represent the 95% bootstrap confidence intervals (CIs). Red-highlighted areas indicate time points where the confidence intervals do not overlap, signifying statistically significant differences between the two groups.

**Figure 6 bioengineering-11-00911-f006:**
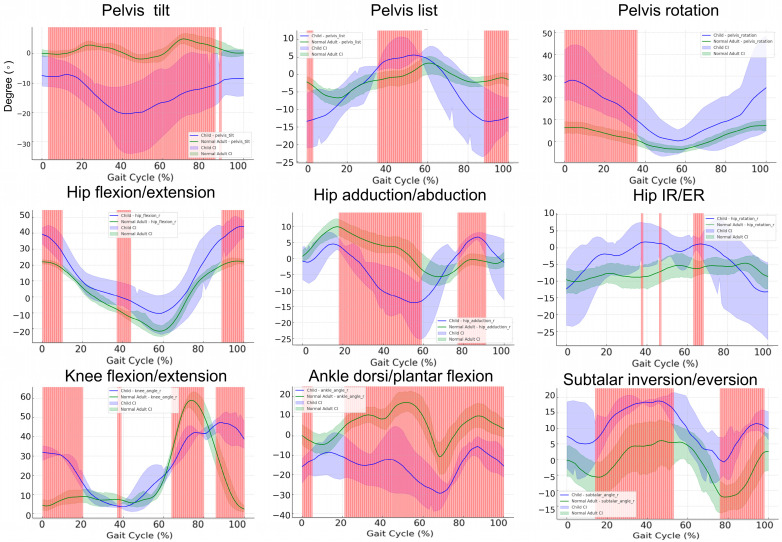
Comparison of bootstrap confidence intervals and non-overlapping regions for hip, knee, and ankle joint kinematics in children and adults. This figure presents a comparison of lower limb joint kinematics between children and healthy adult controls throughout the gait cycle, focusing on the following key parameters: hip flexion/adduction, knee flexion, and ankle plantarflexion/dorsiflexion. The mean joint angle trajectories for children are shown in blue, while those for healthy adults are displayed in green. Shaded regions around each curve represent the 95% bootstrap confidence intervals (CIs). Red-highlighted areas indicate time points where the confidence intervals do not overlap, denoting statistically significant differences between the two groups.

**Table 1 bioengineering-11-00911-t001:** The demographic and clinical characteristics of the control and patient groups.

	Control (*n* = 10) Mean (SD)	Patient (*n* = 10) Mean (SD)	*p*-Value
Age (years)	31.30 (11.55)	51.60 (24.45)	0.034 *
Sex (male/female)	3M/7F	4M/6F	1.000
Height (m)	1.68 (0.09)	1.59 (0.21)	0.230
Weight (kg)	60.50 (16.13)	62.50 (18.91)	0.802
BMI (kg/m^2^)	21.18 (3.83)	24.26 (3.95)	0.093
Gait speed (m/s)	1.10 (0.13)	0.67 (0.31)	0.002 *
Stride length (m)	1.29 (0.15)	0.81 (0.31)	0.001 *
Step width (cm)	12.17 (3.10)	15.58 (3.92)	0.045 *
Cadence (step/min)	104.60 (9.93)	94.70 (28.92)	0.328
Double support (%cycle)	29.35 (2.72)	36.69 (12.50)	0.100
Step length asymmetry (%)	91.23 (11.70)	107.43 (16.76)	0.023 *

* *p*-value < 0.05 is considered statistically significant.

**Table 2 bioengineering-11-00911-t002:** Comparison of peak joint angles between control and patient groups.

Joint	Peak Value (Degree)	Control Mean (SD)	Patients Mean (SD)	*p*-Value
Hip	Flexion	24.225 (2.348)	26.146 (10.074)	0.57
	Extension	−21.736 (5.938)	−11.52 (8.642)	0.007 *
	Adduction	10.599 (5.291)	8.431 (3.068)	0.281
	Abduction	−7.937 (2.402)	−7.544 (2.465)	0.722
	Internal Rotation	−1.208 (3.307)	1.121 (6.209)	0.313
	External Rotation	−13.663 (5.101)	−13.629 (6.114)	0.989
Knee	Flexion	61.492 (5.702)	53.952 (12.767)	0.113
	Extension	1.152 (1.013)	2.168 (2.458)	0.25
Ankle	Dorsiflexion	20.416 (10.514)	16.253 (6.256)	0.299
	Plantarflexion	−14.189 (13.056)	−6.667 (8.03)	0.142
Subtalar	Inversion	17.496 (7.34)	14.758 (8.779)	0.459
	Eversion	−15.709 (6.618)	−11.021 (12.62)	0.316

* *p*-value < 0.05 is considered statistically significant.

## Data Availability

The data presented in this study are available on request from the corresponding author.

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
