# Peer review of "Biomechanical Gait Analysis Using a Smartphone-Based Motion Capture System (OpenCap) in Patients with Neurological Disorders"

_bioengineering, 2024, doi:10.3390/bioengineering11090911_

Round 1

Reviewer 1 Report

Comments and Suggestions for Authors

The sample size is insufficient to establish the scientific validity of the results. Additionally, the non-control group exhibits a high degree of disease variation. In other words, samples from subjects with different diseases were included, which could introduce additional variability in the results. Furthermore, differences in dog breeds could particularly influence hip extension values, potentially affecting the study's findings.

Despite these limitations, the detailed explanation of the technique (except for the application part) and the comprehensive presentation of the results are commendable. The attempt to introduce a new method also has the potential to serve as a reference for future research. However, a few improvements could further strengthen the article:

  1. The introduction section (lines 53-87) would benefit from additional references to support the background information and enhance the context of the study.

  2. Including a few images that demonstrate how iOS devices were used during the application process could help readers better understand the methodology.

  3. The discussion section could be enriched by comparing the study's results with those of other temporospatial gait studies, particularly in terms of application and data acquisition speeds.

Reviewer 2 Report

Comments and Suggestions for Authors

This paper analyses the gait of 20 people: 10 control and 10 with neurological disorders. The analyses show significant differences between both subsets.

This sentence in the abstract is a bit confusing: “These results suggest that OpenCap is a feasible and cost-effective tool for clinical gait analysis, particularly in settings where traditional motion capture systems are impractical.”

Seeing significant differences between both subsets does not mean the system is good. I think the idea is to validate the system using a ground truth (this has already been done in a previous work, isn’t it?) and then, it is used to another application showing significant differences. These differences can be used as biomarkers, and this is (from my point of view) the most interesting aspect.

Comments to improve the paper:

·       At the end of the introduction, I’d suggest including a list of contributions.

·       TRC files: expand initials or include a glossary.

·       Increase font size in all figures.

·       In the evaluation, you provide differences in specific points of the gait cycle. Is it possible to consider metrics comparing the whole cycle?

·       Is it possible to compare the differences obtained using OpenCap with differences in medical reports??

Round 2

Reviewer 2 Report

Comments and Suggestions for Authors

The authors have addressed my comments. Thank you.